# On Distributed Averaging for Stochastic k-PCA

**Aditya Bhaskara**
School of Computing
University of Utah
bhaskara@cs.utah.edu

**Maheshakya Wijewardena**
School of Computing
University of Utah
pmaheshakya4@gmail.com

## Abstract

In the stochastic $k$-PCA problem, we are given i.i.d. samples from an unknown distribution over vectors, and the goal is to compute the top $k$ eigenvalues and eigenvectors of the moment matrix. In the simplest distributed variant, we have $m$ machines each of which receives $n$ samples. Each machine performs some computation and sends an $O(k)$-size summary of the local dataset to a central server. The server performs an aggregation and computes the desired eigenvalues and vectors. The goal is to achieve the same effect as the server computing using $mn$ samples by itself. The main choices in this framework are the choice of the summary, and the method of aggregation. We consider a slight variant of the well-studied *distributed averaging* approach, and prove that this leads to significantly better bounds on the dependence between $n$ and the eigenvalue gaps. Our method can also be applied directly to a setting where the 'right' value of the parameter $k$ (i.e., one for which there is a non-trivial eigenvalue gap) is not known exactly. This is a common issue in practice which prior methods were unable to address.

## 1 Introduction

Principal Component Analysis (PCA) is one of the classic tools for the analysis of high dimensional data. It is used in applications ranging from data visualization, to dimension reduction, to signal de-noising [16, 10]. Formally, the problem is the following: given a collection of data points $x_1, x_2, \ldots, x_n$, the aim is to find a subspace $U$ of dimension precisely $k$ such that captures the most mass of the points. Specifically, the goal is to find a matrix $U(d \times k)$ with orthonormal columns (corresponding to a basis for the desired subspace) to as to maximize $\|\Sigma U\|_F$, where $\Sigma$ is the covariance matrix of the data, defined as $\sum_i x_i x_i^T$. This problem can be solved efficiently by computing the singular value decomposition (SVD) (see [9]).

In the stochastic version of the problem, the data is viewed as samples from an unknown distribution $\mathcal{D}$ over points in $\mathbb{R}^d$, and the goal is to find the top $k$ singular directions of the distribution covariance matrix (or the *second moment* matrix) $\Sigma = \mathbb{E}_{x \sim \mathcal{D}} x x^T$. The question of how many samples from $\mathcal{D}$ are needed to find a good estimate for the $k$-PCA is extensively studied ([15, 2, 20], and tight bounds that involve the gap between the $k$th and the $(k+1)$th eigenvalues of $\sigma$ can be obtained using matrix concentration inequalities [1, 22].

In this paper, we consider distributed algorithms for stochastic PCA, where the samples from $\mathcal{D}$ are distributed across machines, and the goal is to use a small amount of communication and find a solution that approximates the PCA of the distribution. Our focus will be on the simplest model, where we have $m$ machines that each has access to $n$ i.i.d. samples of the data. Each machine sends one *summary* to a central server. The server, using the summaries from the different machines, computes the estimate of the PCA (this will be known as the *aggregation* step).

This distributed procedure is well-studied for various optimization problems [14, 24, 25]. The most well-known example is distributed convex optimization, where the goal is to optimize the loss

$L(\theta) = \mathbb{E}_{x \in \mathcal{D}} f(x, \theta)$ for some convex function $f$. Here, it turns out that a simple procedure known as *distributed averaging* yields good guarantees. Machines simply optimize the objective on their local dataset and send the solution $\theta$ to the server, and the central server averages the local solutions. Tight bounds are known for distributed averaging for various convex objectives (see [24]).

A natural problem not covered by the general results on convex optimization is PCA. Garber et al. [8] studied the power of distributed averaging for PCA. They showed that for the problem of computing the top eigenvector, simply averaging the best vectors for the different machines does *not* work (the issue being one of having the right signs). However, it turns out that averaging with appropriately chosen signs works well, as long as there is a *sufficient gap* between $\lambda_1$ and $\lambda_2$. Fan et al. [7] extended this idea to the case of finding the top $k$ principal subspaces of the covariance matrix. They show that as long as every machine has sufficiently many samples (a quantity that depends on the gap between $\lambda_k$ and $\lambda_{k+1}$), distributed averaging of the projection matrices output by different machines (followed by a $k$-SVD) yields a good approximation.

## 1.1 Problem setup and motivation

Let us start with some basic notation. For a real symmetric matrix $M$, we use $\lambda_j(M)$ to refer to its $j$-th largest eigenvalue. We denote by $M_k$ the best $k$-rank approximation of $M$. Also, the trace of $M$ will be denoted as $\mathrm{Tr}(M) = \sum_j \lambda_j(M)$. The Frobenius norm is $\|M\|_F := \sqrt{\sum_j \lambda_j(M)^2}$. For a $r \in [d]$, we define $\Delta_r$ to be the eigenvalue gap $\lambda_r - \lambda_{r+1}$.

**Formal setting for distributed stochastic PCA.** Let $\mathcal{D}$ be an (unknown) sub-Gaussian distribution over vectors in $\mathbb{R}^d$.[1] We have $m$ machines, each of which receives $n$ i.i.d. samples from $\mathcal{D}$. $A$ denotes the covariance matrix of the distribution $\mathcal{D}$, i.e., $A = \mathbb{E}_{x \in \mathcal{D}} x x^T$. Let the spectral decomposition of $A$ be denoted $U \Lambda U^T$ where $U \in \mathbb{R}^{d \times d}$ is a matrix with orthonormal columns and $\Sigma = \mathrm{diag}(\lambda_1, \lambda_2, \ldots, \lambda_d)$. The aim is to find the vectors $u_1, u_2, \ldots, u_k$ (the first $k$ columns of $U$) and the corresponding eigenvalues $\lambda_1, \lambda_2, \ldots, \lambda_k$.

**Motivation.** The works [7, 8] have two key limitations. First, they are aimed at finding the $k$-PCA subspace, for a given $k$. Even modifying the goal slightly, e.g., requiring the algorithm to output each of the top $k$ PCA directions individually, requires more communication, and a sample complexity (per machine) that has a quadratic dependence on the individual gaps, as we explain below. Second, and more importantly, these works assume the knowledge of a number $k$ for which there exists an eigenvalue gap. This is quite unrealistic in practice. Can we design algorithms that can work with only a rough idea of the location of the gap? Our main contribution is to handle these two issues. We provide novel estimation bounds and validate them using experiments on real and synthetic data.

The first restriction above is quite serious at a quantitative level. Ignoring other terms, the works of [7, 8] require (in order to estimate the $k$-subspace), a value of $n \geq 1/\Delta_k^2$. Intuitively, this corresponds to a requirement of each machine having a *rough* estimate of the top-$k$ PCA subspace. Their main results then can be interpreted as saying that under this assumption, the server can obtain a significantly better estimate of the top-$k$ subspace than the individual machines. Further, the estimation errors are only obtained for the matrix $U_k U_k^T$ (i.e., the projection matrix to the top-$k$ subspace). If one needs each of the top $k$ singular directions, the procedure needs to be re-done for each index (and this requires having $n \geq 1/\Delta_{\min}^2$, where $\Delta_{\min} = \min_{i \in k} \Delta_i$, which could be tiny).

Note that our setting is slightly different from the deterministic case of the *distributed PCA*, where we have a matrix $U$ whose columns are *arbitrarily* distributed across machines, and the goal is to find the best $k$-subspace. Moreover, the objective is not always to find the right eigenvalues/vectors, but to approximate the value of the low-rank error (see [4, 11] and references therein). These results extend the works of [19, 5] where the power of subspace embeddings in matrix approximation is shown to distributed settings. In this case, in order to obtain a $1 + \epsilon$ low rank error in Frobenius norm for $A$, each machine needs to communicate $O(k/\epsilon)$ vectors. It is evident from their work that the sketching methods perform better as the sketch size grows. In contrast to these results, a part of our goal is to discuss the trade-off between $n$ and the quality of the approximations. Applying these sketching methods in our setting will yield error terms that depend only on the sketch size and cannot be controlled by $n$ or $m$, thus making them undesirable for this setting.

## 1.2 Our contributions

In Theorem 1 and its corollaries (Section 2.1), we show that as long as $n \geq \Omega(1/\Delta_k^2)$, we can (using a "single sketch") compute estimates $\widetilde{v}_i$ of each of the vectors $v_1, v_2, \ldots, v_k$ up to an error

$$\|\widetilde{v}_i - v_i\| \leq \frac{1}{\delta_i} \cdot \left( \frac{\kappa_1}{n} + \frac{\kappa_2}{\sqrt{mn}} \right), \text{ where } \delta_i := \min(\lambda_{i-1} - \lambda_i, \lambda_i - \lambda_{i+1}),$$

where $\kappa_1, \kappa_2$ are factors that are not dominant when $\Delta_k$ is large enough. Further, the amount of communication per machine is $O(kd)$, i.e., each machine communicates $k$ vectors in $\mathbb{R}^d$.

*Remark.* Note that if each individual machine is to achieve $\|\widetilde{v}_i - v_i\| \leq 1/4$, the above requirement translates to $\min(n, \sqrt{mn}) \geq 4/\delta_i$. To achieve this error using prior work, one needs $n \geq \Omega(1/\delta_i^2)$. Our result can be a significant improvement as $m$ grows. Specifically, in our setting the individual machines need not be able to obtain any estimate of $v_i$, but the corresponding average is still accurate.

Our next result (Theorem 9) is in a setting in which we only approximately know the location of the gap in the eigenvalues (as is common in practice). In particular, suppose that there exists a $k \in (k_0, k_1)$ such that $\Delta_k$ is large enough. Then, using a single sketch of $k_1$ vectors (i.e. space $O(k_1 d)$, we show an efficient way to find $k$, and thus also achieve the same guarantees as our first result (described above).

Using prior results (at least directly) to solve this problem lead to two issues. First, we need to run the averaging procedures for each $k$ in the range $(k_0, k_1)$. And more importantly, it is not clear how to determine which of the PCA directions obtained are accurate and which are not (because accuracy guarantees depend on the consecutive gaps, and we do not know which gap is large).

Finally, we run experiments on both real and synthetic datasets (the latter gives us a way to control the eigenvalue gaps), and establish that our theoretical bounds are reflected accurately in practice.

## 2 Spectral approximation via distributed averaging

We start by introducing notation that we will use for the rest of the paper and stating the theorems formally. Recall that the $j$th machine gets $n$ i.i.d. vectors from a sub-Gaussian distribution $\mathcal{D}$, and let $\widehat{A}^{(j)}$ denote the empirical covariance matrix. Also, for the $j$th machine, let $\widehat{A}_k^{(j)}$ denote the best rank $k$ approximation of $\widehat{A}^{(j)}$, and denote its SVD by $\widehat{U}_k^{(j)} \widehat{\Lambda}_k^{(j)} (\widehat{U}_k^{(j)})^T$. We also define $\widehat{V}_k^{(j)} = \widehat{U}_k^{(j)} (\widehat{\Lambda}_k^{(j)})^{1/2}$. The columns of this matrix are what each machine sends to the central server.

We also define the average across machines: $\widetilde{A}_k = \frac{1}{m} \sum_{j \in [m]} \widehat{A}_k^{(j)}$.

---

**Algorithm 1** Distributed Averaging (parameter $k$)

**Local:** On each machine, compute the rank-$k$ SVD of the empirical covariance matrix $\widehat{A}^{(j)}$, and send $\widehat{V}_k^{(j)}$ (as defined above) to the server.

**Server:** On the central server, compute $\widetilde{A}_k = \frac{1}{m} \sum_{j=1}^{m} \hat{V}^{(j)} (\hat{V}^{(j)})^T$. Then output the top $k$ eigenvalues and the corresponding eigenvectors of $\widetilde{A}_k$.

---

The procedure above differs from the prior works [8, 7] in the choice of the summary (i.e., what the individual machines send to the central server). Algorithm 1 uses the eigenvectors weighted by the square root of the eigenvalues, while unweighted vectors are used in the prior work. This turns out to give us three advantages: (i) an efficient way to obtain the individual eigenvectors $v_1, v_2, \ldots, v_k$, (ii) use the summary for different values of $k$, as we will see in Section 4, and also (iii) improved bounds on the parameters $m, n$ (especially in experiments).

### 2.1 Guarantees for eigenvalue and eigenvector estimation

We now formally state the guarantees obtained by the procedure above in order to estimate the eigenvalues and eigenvectors of $A$. We start with a general theorem about approximating the best $k$-rank approximation of $A$: $A_k$, which will imply both of these statements.

**Theorem 1.** *There exist constants $C_1$ and $C_2$ such that*

$$\left\| \|\widetilde{A}_k - A_k\|_F \right\|_{\psi_1} \leq C_1 \frac{\kappa_1}{n} + C_2 \frac{\kappa_2}{\sqrt{mn}},$$

*where $\kappa_1 = \sqrt{k}\lambda_1^2 \cdot Tr(A)/\Delta_k^2$ and $\kappa_2 = \lambda_1\sqrt{k\lambda_1 \cdot Tr(A)}/\Delta_k$*

The statement uses subgaussian norms described in section 1.1. By definition, we can rephrase the theorem as a concentration bound: the probability of $\|\widetilde{A}_k - A_k\|_F$ exceeding $\log(1/\delta)$ times the RHS is at most $\delta$. Using known perturbation bounds, we can show the following corollaries.

**Corollary 2.** *For all $i \leq k$, there exist constants $C_1$ and $C_2$ such that*

$$\left\| |\lambda_i(\widetilde{A}_k) - \lambda_i(A)| \right\|_{\psi_1} \leq C_1 \frac{\kappa_1}{n} + C_2 \frac{\kappa_2}{\sqrt{mn}}.$$

The corollary follows from Theorem 1 using Weyl's inequality [21].

**Corollary 3.** *Define $\delta_i = \min\{(\lambda_i - \lambda_{i-1}), (\lambda_{i+1} - \lambda_i)\}$.[2] For $i \leq k$, there exist constants $C_1$ and $C_2$ such that*

$$\left\| 1 - (\widetilde{u}_i^T u_i)^2 \right\|_{\psi_1} \leq C_1 \frac{\kappa_1^2}{\delta_i^2 n^2} + C_2 \frac{\kappa_2^2}{\delta_i^2 mn},$$

*where $\widetilde{u}_i$ is the eigenvector corresponding to the $i$th largest eigenvalue of $\widetilde{A}_k$.*

The proof follows from the Davis-Kahan sin-$\Theta$ theorem [21].

As outlined in Section 1.2, when the gap $\delta_i \ll \Delta_k$, using the summary corresponding to $k$ has a significant advantage over using the one for $i$. This results in better guarantee (compared to the procedures of [8, 7]) when recovering $u_i$, for $1 \leq i \leq k$.

## 3 Analysis: estimating the rank-$k$ approximation of $A$

Our goal in this section will be to show that $\widetilde{A}_k$ approximates $A_k$ accurately, thereby proving Theorem 1.

**Outline of the argument.** The key step is to define the matrix $A^*$, which is the expectation of $\widehat{A}_k^{(j)}$. As all the machines receive inputs drawn from $\mathcal{D}$, this is independent of $j$. The argument proceeds in two steps, similar to the works of [7] and [8]. The first step is showing that $\|A^* - A_k\|$ (in other words, the *bias*) is small. This is the harder step, and involves showing that one obtains non-trivial cancellations. In other words, even though $\|\widehat{A}_k^{(j)} - A_k\|$ is of the order $O(1/\sqrt{n})$, we will show that $\|A^* - A_k\|$ is of the order $O(1/n)$. The second step is to show that $\widetilde{A}_k$, which is the empirical average of $\widehat{A}_k^{(j)}$ over the $m$ machines, is close to $A^*$. This is proved using a matrix concentration bound, originally due to [3].

To summarize, let us define (noting that the RHS is independent of $j$),

$$A^* = \mathbb{E}[\widehat{A}_k^{(j)}].$$

We have $\|\widetilde{A}_k - A_k\| \leq \|A^* - A_k\| + \|\widetilde{A}_k - A^*\|$. The first term will be referred to as the bias and the second as the variance. In what follows we will bound the terms separately.

### 3.1 Analyzing the bias term

We now show the following theorem about the bias term.

**Theorem 4.** *There is a constant $C$ such that*

$$\|A^* - A_k\|_F \leq C \frac{\sqrt{k}\lambda_1^2 \cdot Tr(A)}{\Delta_k^2 n}.$$

In what follows, we abuse notation slightly and denote $\widehat{A} = \widehat{A}^{(j)}$ for some machine $j$. As we are finally interested in the expectation, the choice of $j$ will not matter. Define $\widehat{A} = A + E$ and let $\epsilon = \|E\|_2/\Delta_k$. By definition, $A = \mathbb{E}[\widehat{A}]$. Let us also define the projection matrices $\Pi = U_k U_k^T$ and $\widehat{\Pi} = \widehat{U}_k \widehat{U}_k^T$.

The main idea behind the proof of theorem 4 is we express $\widehat{A}_k - A_k$ in a single machine using linear and quadratic terms of $E$. Once we consider the expectation of this error, the linear terms of $E$ ($O(1/\sqrt{n})$ which is dominant in magnitude) will become zero, thus giving the bound for $O(1/n)$ error bias in theorem 4. The first lemma gives a coarse bound, which we will use when $\|E\|$ is large.

**Lemma 5.** *Let $\widehat{A}_k$ be the rank-$k$ approximation on one of the machines, and let $E$ be as defined above. Then*

$$\widehat{A}_k - A_k = \Pi E + H, \text{ where } \|H\|_F \leq 2\sqrt{k}\frac{\lambda_1 \|E\|_2}{\Delta_k} + 2\sqrt{k}\frac{\|E\|_2^2}{\Delta_k}.$$

The next lemma shows that when $\epsilon = \|E\|_2/\Delta_k$ is small, we have a much better bound.

**Lemma 6.** *Let $\widehat{A}$ satisfy the condition $\epsilon = \|A - \widehat{A}\|_2/\Delta_k \leq 1/10$. There exists a linear function $f : \mathbb{R}^{d \times k} \mapsto \mathbb{R}^{d \times k}$ and a constant $C$ such that*

$$\widehat{A}_k - A_k = \Pi E + \left(f(EU_k)U_k^T + U_k f(EU_k)^T\right) A + H, \text{ where } \|H\|_F \leq \frac{c\sqrt{k}\|E\|_2^2(\lambda_1 + \|E\|_2)}{\Delta_k^2}.$$

The lemma is a consequence of a result in [7] showing that in this case, $\widehat{\Pi}$ has a sufficiently good first order approximation in terms of $E$.

*Proof.* Lemma 2 of [7] shows that

$$\widehat{\Pi} = \Pi + f(EU_k)U_k^T + U_k^T f(EU_k)^T + E',$$

where (a) $f$ is a linear function as in the statement of the theorem that also satisfies $\|f(.)\|_F \leq \|.\|_F/\Delta_k$, and (b) $E'$ is a matrix with $\|E'\|_F \leq 24\sqrt{k}\|E\|_2^2/\Delta_k^2$ (this is only true under the assumption we have, i.e., $\epsilon \leq 1/10$). Using this,

$$\begin{aligned}
\widehat{A}_k - A_k &= \widehat{\Pi}(A + E) - A_k \\
&= \left(\Pi + f(EU_k)U_k^T + U_k f(EU_k)^T + E'\right)(A + E) - A_k \\
&= \Pi E + \left(f(EU_k)U_k^T + U_k f(EU_k)^T\right) A + \left(f(EU_k)U_k^T + U_k^T f(EU_k)^T\right) E + E'A + E'E
\end{aligned}$$

Thus to show the lemma, the error term is

$$H = \left(f(EU_k)U_k^T + U_k^T f(EU_k)^T\right) E + E'A + E'E.$$

To bound the first term, note that $\|f(EU_k)\|_F \leq \frac{\|EU_k\|_F}{\Delta_k} \leq \frac{\sqrt{k}\|E\|_2}{\Delta_k}$. Thus we have

$$\left\|\left(f(EU_k)U_k^T + U_k^T f(EU_k)^T\right) E\right\|_F \leq \|f(EU_k)U_k^T + U_k^T f(EU_k)^T\|_F \|E\|_2 \leq \frac{2\sqrt{k}\|E\|_2^2}{\Delta_k}.$$

The second term can be bounded (using the bound on $\|E'\|$ above), by $24\sqrt{k}\lambda_1\|E\|_2^2/\Delta_k^2$. Using the bound on $E'$ again completes the proof of the lemma. $\square$

Note that the two lemmas give different linear approximations of $\widehat{A}_k - A_k$. However, in order to take expectation, we need the same function. Luckily, we observe that the one from Lemma 6 can be used in the place of one from before, with small error. To this end, note that

$$\left\|\left(f(EU_k)U_k^T + U_k f(EU_k)^T\right) A\right\|_F \leq 2\|f(EU_k)\|_F \|A\|_2 \leq \frac{2\sqrt{k}\lambda_1\|E\|_2}{\Delta_k}, \qquad (1)$$

from the property of $f$ mentioned earlier (shown in [7]).

We can now prove Theorem 4.

*Proof of Theorem 4.* Using the observation in (1), Lemma 5 implies that for all $E$, we have

$$\widehat{A}_k - A_k = \Pi E + \left(f(EU_k)U_k^T + U_k f(EU_k)^T\right) A + H, \text{ where } \|H\|_F \leq 4\sqrt{k}\frac{\|E\|_2(\lambda_1 + \|E\|_2)}{\Delta_k}. \tag{2}$$

Using this expression for bounding $\|E\|_F$ when $\epsilon \geq 1/10$, and the one from Lemma 6 when $\epsilon$ is smaller, we can now take the expected value of $\widehat{A}_k - A_k$. The linear terms in $E$ will evaluate to zero. Thus we have

$$\|\mathbb{E}[\widehat{A}_k - A_k]\|_F \leq \mathbb{E}[Q_1 \mid \epsilon \geq 1/10] + \mathbb{E}[Q_2 \mid \epsilon \geq 1/10],$$

where $Q_1$ and $Q_2$ are bounds on $\|H\|_F$ from (2) and Lemma 6 respectively. Now, conditioned on $\|E\|_2/\Delta_k \geq 1/10$, it is trivially true that $\|E\|_2/\Delta_k \leq 10\|E\|_2^2/\Delta_k^2$. Thus we can simplify the above as

$$\|\mathbb{E}[\widehat{A}_k - A_k]\|_F \leq \mathbb{E}\left[\frac{C\sqrt{k}\|E\|_2^2(\lambda_1 + \|E\|)}{\Delta_k^2}\right].$$

Using the subgaussian property of the moments of our distribution, we have that the expectation above is dominated by $\mathbb{E}[\|E\|_2^2]$ term (due to the multiplier $\lambda_1$). This gives

$$\|\mathbb{E}[\widehat{A}_k - A_k]\|_F \leq \frac{C\sqrt{k}\lambda_1^2 \cdot \text{Tr}(A)}{n\Delta_k^2}.$$

This completes the proof of the theorem. $\qquad\square$

## 3.2 Analyzing the variance term

We now need to show that the average of the matrices $\widehat{A}_k$ is close to the expectation (which is $A^*$). The main idea is to use the concentration inequality due to Bosq [3] (see also Lemma 4 of [7]). The inequality lets us bound the $\psi_1$ norm of the average of i.i.d. random variables using the $\psi_1$ norm of the individual variables.

To this end, we first show the following.

**Lemma 7.** *Suppose each machine receives $n$ points, where $n \geq \frac{\lambda_1 Tr(A)}{\Delta_k^2}$. Then, there is a constant $C$ such that*

$$\left\|\|\widehat{A}_k - A^*\|_F\right\|_{\psi_1} \leq C\frac{\lambda_1}{\Delta_k}\sqrt{\frac{k\lambda_1 \cdot Tr(A)}{n}}.$$

Now we analyze the average of $\widehat{A}_k$.

**Theorem 8.** *There exists a constant $C$ such that the matrix $\widetilde{A}_k$, i.e. the average of the matrices $\widehat{A}_k^{(i)}$, satisfies*

$$\left\|\|\widetilde{A}_k - A^*\|_F\right\|_{\psi_1} \leq C\frac{\lambda_1}{\Delta_k}\sqrt{\frac{k\lambda_1 Tr(A)}{mn}}.$$

Theorems 4 and 8 together complete the proof of our main approximation result, Theorem 1. As observed in Section 2.1, this also completes the proofs of Corollaries 2 and 3.

## 4 Algorithm for imprecise $k$

We now consider the setting in which we do not exactly know the value of $k$ for which $\lambda_k - \lambda_{k+1}$ is "large". Knowing that some $k$ in the interval $(k_0, k_1)$ satisfies an appropriate gap assumption, we will give an algorithm that can, using $O(k_1)$ columns of communication per machine, (a) find such a $k$, and (b) compute all the eigenvalues and eigenvectors with guarantees matching ones from the case in which we know $k$ (i.e. Theorem 1).

Our algorithm relies on the following theorem. As defined earlier, for any $t \geq 1$, denote $\widetilde{A}_t := \frac{1}{m}\sum_i \widehat{A}_t^{(i)}$ (i.e., the empirical average of the rank-$t$ approximations on the individual machines).

Now the main advantage of having every machine sending across the eigenvectors $\widehat{v}_i$ scaled by $\sqrt{\widehat{\lambda}_i}$ (Algorithm 1) is that if a machine sends this information for $1 \leq i \leq k_1$, then the central server can compute $\widetilde{A}_t$ for *every* $1 \leq t \leq k_1$.

**Theorem 9.** *Let $\widetilde{A}_t$ be defined as above, and let $\delta > 0$ be a given parameter. Let $k$ be an integer for which $\Delta_k = \lambda_k - \lambda_{k+1}$ is sufficiently large, in particular, so that for the given $m, n, \delta$, we have*

$$\Delta_k \geq C \left( \frac{\kappa_1}{n} + \frac{\kappa_2}{\sqrt{mn}} \right) \log(1/\delta),$$

*where $\kappa_1$ and $\kappa_2$ are as defined in Theorem 1, and $C$ is an appropriate constant. Then with probability at least $1 - \delta$, for all $t \geq k$, the matrix $\widetilde{A}_t$ has its top $k + 1$ eigenvalues $\theta_1 \geq \theta_2 \geq \cdots \geq \theta_k \geq \theta_{k+1}$ that satisfy*

$$|\theta_i - \lambda_i| \leq O \left( \frac{\kappa_1}{n} + \frac{\kappa_2}{\sqrt{mn}} \right) \log(1/\delta), \text{ for } 1 \leq i \leq k, \text{ and} \tag{3}$$

$$\theta_{k+1} \leq \lambda_{k+1} + O \left( \frac{\kappa_1}{n} + \frac{\kappa_2}{\sqrt{mn}} \right) \log(1/\delta). \tag{4}$$

Proof of this theorem is deferred to Section A.4 of the supplementary material.

**Estimating the location of the gap.** The theorem shows that one can use *any* value of $t \geq k$ in order to estimate all the eigenvalues up to $k$, and also the gap between $k$ and $k + 1$. Thus, if we only know the approximate location of a gap (some $k$ between $k_0$ and $k_1$, we can use Algorithm 1 with $k = t_1$, and using the above result, find the $k$ with the desired gap. Knowing $k$, the server can then compute $\widetilde{A}_k$ (using only the information it has), and this leads to a finer estimate of the matrix $A_k$.

**Estimating the eigenvectors.** The theorem above can also be used (together with the sin-$\Theta$ theorem) to show estimates on eigenvector estimation (as in Corollary 3). While the bounds are qualitatively similar to those in Corollary 3, we observe that in practice, using $t > k$ is significantly better for approximating the top $k$ eigenspace to a good accuracy.

## 5 Experiments

We validate our results with experiments using synthetic and real datasets. We simulated a distributed environment on a single machine.

### 5.1 Synthetic dataset

We generated vectors in $\mathbb{R}^d$ from a multivariate-Gaussian distribution with mean $\mathbf{0}$ and the covariance matrix $A = U \Lambda U^T$. $\Lambda(1,1) = 1$, $\Lambda(i,i) = 0.9\Lambda(i-1,i-1)$ for $i = 2, \ldots, 6$. We set $\Lambda(7,7) = \Lambda(6,6) - 0.3$. For $7 < i \leq 50$, we set $\Lambda(i,i) = 0.9\Lambda(i-1,i-1)$. So there is a gap of $\Delta_6 = 0.3$. In this experiment we fixed the number of machines to $m = 50$. We computed top 3 eigenvectors using the algorithm 1 increasing the number of points $n$ per machine and compared them with the top 3 eigenvectors of the population covariance matrix. Note that it is not possible to use prior methods for computing individual eigenvectors. We first computed the eigenvectors by communicating only the top $k = 3$ weighted vectors. We then compute that by communicating $k = 7$ weighted vectors so that it includes the eigengap $\Delta_6$. We computed the error $(1 - (u_i^T \tilde{u}_i)^2, i = 1, 2, 3)$ for both these cases. These results are averaged over 200 iterations

As observed in Figure 1, these results are consistent with our theoretical bounds for the case where correct eigengap is not known (but it is located within the $k$ we communicate).

### 5.2 Real datasets

We used 3 real datasets to evaluate our methods (Table 1)[ [13, 17, 6]]. Each dataset has $N$ points and $d$ features. In these experiments we compute the error of the subspace spanned by the top $r$ eigenvectors ($\|U_r U_r^T - \tilde{U}_r \tilde{U}_r^T\|_F$). We consider each dataset $X$ as the population matrix, then the population covariance matrix $A = XX^T$. In each machine we sampled $n$ columns from these $X$ matrices uniformly at random. For these experiments we fixed number of machines $m = 50$. Each result is averaged over 200 iterations.

We compare the prior method by [7] (unweighted) with our methods. In one of the cases we communicate exactly $r$ vectors (weighted $r$) and in the other case we communicate a slightly higher

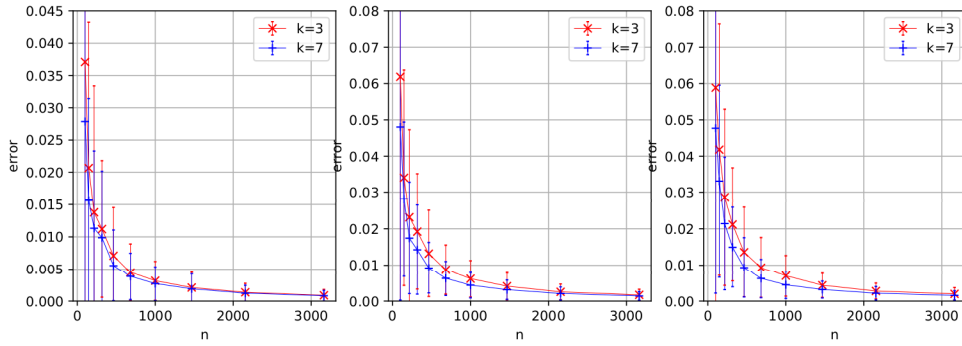

Figure 1: Estimation errors of first eigenvector (left), second eigenvector (middle), and third eigenvector (right) for $k = 3$ and $k = 7$ vs. samples size $n$ per machine.

| Dataset | N | d | r | t |
|---|---|---|---|---|
| MNIST-small | 20000 | 196 | 5 | 15 |
| NIPS-papers | 11463 | 150 | 5 | 15 |
| FMA-music | 21314 | 518 | 10 | 70 |

Table 1: Dataset information

($t > r$) number of vectors (weighted $t$). This is towards the ends of demonstrating our theoretical results for the case where we do not know the exact location of a reasonable eigengap. Note that it is not possible to compute the correct eigenspace using prior methods if we do not communicate the exact number of required vectors.

Similar to the synthetic dataset experiments we computed the error of each method varying the sample size $n$ per machine (Figure 2).

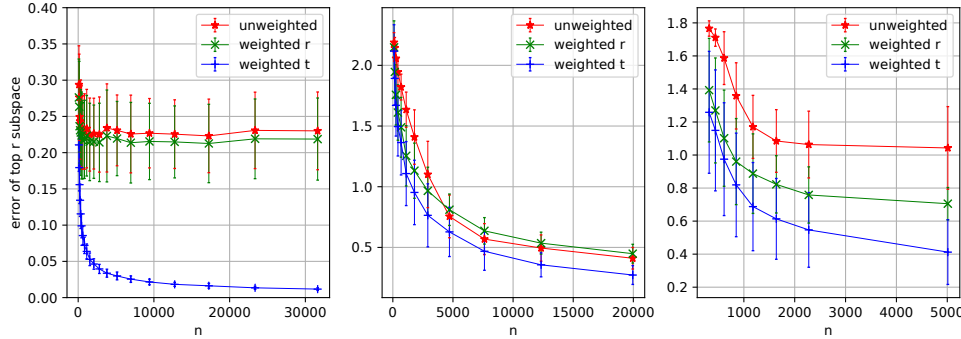

Figure 2: Estimation errors of top $r$ subspace of MNIST-small dataset (left), NIPS-papers dataset (middle), FMA-music dataset (right) vs. unweighted, weighted $r$, weighted $t$ averaging.

## Footnotes

[1] As in the prior works [7, 8], the data distribution is assumed to have sub-Gaussian tails ([12, 18]), i.e., there exists a constant $C > 0$ such that $\|(u^T x)^2\|_{\psi_1} \leq C \mathbb{E}[(u^T x)^2]$, $\forall u \in \mathbb{R}^d$. The $\psi_1$ norm of a random variable $X$ is $\|X\|_{\psi_1} = \sup_{p \geq 1} (\mathbb{E}|X|^p)^{1/p}/p$ (see [23]).

[2]To deal with the border cases $i = 1, d$, define $\lambda_0 = +\infty$ and $\lambda_{d+1} = -\infty$.

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
