[Supplementary Material]


# A  Supplementary material: missing proofs

## A.1  Proof of Lemma 5

*Proof.* From the Davis-Kahan sin $\Theta$ theorem, $\|\widehat{\Pi} - \Pi\|_2 \leq 2\|E\|_2/\Delta_k$ therefore $\|\widehat{\Pi} - \Pi\|_F \leq 2\sqrt{k}\|E\|_2/\Delta_k$. This implies that $\widehat{\Pi} = \Pi + E'$, where $E' \in \mathbb{R}^{d \times d}$ and $\|E'\|_F \leq 2\sqrt{k}\|E\|_2/\Delta_k$. Then

$$
\begin{aligned}
\widehat{A}_k - A_k &= \widehat{\Pi}(A + E) - A_k \\
&= (\Pi + E')(A + E) - A_k \\
&= \Pi E + E'A + E'E
\end{aligned}
$$

Third equation is from the fact that $\Pi A = A_k$. Thus the $H$ in the statement can be set to $E'A + E'E$. Let us bound its norm. We have $\|E'A\|_F \leq \|E'\|_F\|A\|_2 = \lambda_1\|E'\|_F$.[3] The next term can similarly be bounded by $\|E'\|_F\|E\|_2$. Combining these implies the claim. □

## A.2  Proof of Lemma 7

*Proof.* The proof requires relating $A_k$, because it is easier to obtain a bound on $\left\|\|\widehat{A}_k - A_k\|_F\right\|_{\psi_1}$.

Let us write $\widehat{A}_k - A^*$ as $(\widehat{A}_k - A_k) + (A_k - A^*)$. As in the proof of Lemma 5,

$$
\begin{aligned}
\|\widehat{A}_k - A_k\|_F &= \|\widehat{\Pi}(A + E) - \Pi A\|_F \\
&\leq \|(\widehat{\Pi} - \Pi)A\|_F + \|\widehat{\Pi}E\|_F \\
&\leq \|\widehat{\Pi} - \Pi\|_F \lambda_1 + \sqrt{k}\|E\|_2 \leq \frac{2\sqrt{k}\lambda_1\|E\|_2}{\Delta_k} + \sqrt{k}\|E\|_2 \leq \frac{3\sqrt{k}\lambda_1\|E\|_2}{\Delta_k}.
\end{aligned}
$$

Now for the second term, using triangle inequality,

$$
\begin{aligned}
\|A^* - A_k\|_F &= \|\mathbb{E}[\widehat{A}_k - A_k]\|_F \leq \mathbb{E}[\|\widehat{A}_k - A_k\|_F] \\
&\leq \left\|\|\widehat{A}_k - A_k\|_F\right\|_{\psi_1} \leq \frac{3\sqrt{k}\lambda_1\left\|\|E\|_2\right\|_{\psi_1}}{\Delta_k}.
\end{aligned}
$$

Thus we have

$$
\left\|\|\widehat{A}_k - A^*\|_F\right\|_{\psi_1} \leq \left\|\|\widehat{A}_k - A_k\|_F\right\|_{\psi_1} + \|A^* - A_k\|_F \leq \frac{6\sqrt{k}\lambda_1\left\|\|E\|_2\right\|_{\psi_1}}{\Delta_k}.
$$

Using Lemma 3 from [6] now completes the proof. □

## A.3  Proof of Theorem 8

*Proof.*

$$
\|\widetilde{A}_k - A^*\|_F = \|\frac{1}{m}\sum_{i=1}^{m} \widehat{A}_k^{(i)} - A^*\|_F
$$

$\widehat{A}_k^{(i)} - A^* \in \mathbb{R}^{d \times d}$. Let us define $Y_i$ as a $\mathbb{R}^{d^2}$ vector which is equal to the flattened $\widehat{A}_k^{(i)} - A^*$ matrix. Now $\|\frac{1}{m}\sum_{i=1}^{m} Y_i\| = \|\frac{1}{m}\sum_{i=1}^{m} \widehat{A}_k^{(i)} - A^*\|_F$. $\mathbb{E}[Y_i] = 0$ and $\left\|\|Y_i\|\right\|_{\psi_1} \leq C_1 \frac{\lambda_1}{\Delta_k}\sqrt{\frac{k\lambda_1 Tr(A)}{n}}$ for a constant $C_1$. Using Lemma 4 in [6] (which is a consequence of Theorem 2.5 of [3]), for a constant

324    $C_2$

$$\left\|\,\|\frac{1}{m}\sum_{i=1}^{m}Y_i\|\,\right\|_{\psi_1} = \left\|\,\|\sum_{i=1}^{m}\frac{Y_i}{m}\|\,\right\|_{\psi_1}$$

$$\leq \sqrt{\sum_{i=1}^{m}\frac{1}{m^2}C_1^2\frac{\lambda_1^2}{\Delta_k^2}\cdot\frac{k\lambda_1 Tr(A)}{n}}$$

$$\leq C_2\frac{\lambda_1}{\Delta_k}\sqrt{\frac{k\lambda_1 Tr(A)}{mn}}$$

325    This completes the proof.      $\square$

### A.4    Proof of Theorem 9

327 *Proof.* Let us define $\widetilde{B}_k = \frac{1}{m}\sum_{i\in[m]}\widehat{A}^{(i)} - \widehat{A}_k^{(i)}$. By definition, $\widetilde{B}_k = \widetilde{A} - \widetilde{A}_k$, where $\widetilde{A}$ is simply
328 $\frac{1}{m}\sum_{i\in[m]}\widehat{A}^{(i)}$. We start by showing some basic properties about $\widetilde{A}$, $\widetilde{A}_k$ and $\widetilde{B}_k$.

329 First, note that $\widetilde{A}$ is the empirical average (over $m$ machines) of $\widetilde{A}^{(i)}$, and each such matrix is the
330 empirical average (over $n$) samples of $xx^T$. Since samples across and within machines are all i.i.d.,
331 the difference $A - \widetilde{A}$ is simply the error in the estimate of $A$ using $mn$ i.i.d. samples $x \sim \mathcal{D}$. Thus,
332 using Lemma 3 of [6], we have

$$\left\|\,\|A - \widetilde{A}\|_2\,\right\|_{\psi_1} \leq C\sqrt{\frac{\lambda_1\mathrm{Tr}(A)}{mn}}. \tag{5}$$

333 From Theorem 1, we have that for any $\delta > 0$, with probability at least $1 - \delta$,

$$\|A_k - \widetilde{A}_k\|_F \leq \left(\frac{\kappa_1}{n} + \frac{\kappa_2}{\sqrt{mn}}\right)\log(1/\delta). \tag{6}$$

334 Let $\Pi$ denote the projection matrix onto the span of the top $k$ SVD directions of $A$, and $\Pi^\perp = I - \Pi$.
335 We will also denote $\kappa = \frac{\kappa_1}{n} + \frac{\kappa_2}{\sqrt{mn}}$, for convenience.

336 Next, we claim that $\|\Pi\widetilde{B}_k\|$ is $O(\kappa\log(1/\delta))$ with high probability. To see this, write $\widetilde{B}_k = \widetilde{A} - \widetilde{A}_k =$
337 $(A - A_k) + (A_k - \widetilde{A}_k) - (A - \widetilde{A})$. Now, $\Pi(A - A_k) = 0$, by definition. Thus, using (6) and (5),
338 the claim follows.

339 Note that our goal is not to reason about the eigenvalues of $\widetilde{A}_k$, but the eigenvalues of $\widetilde{A}_t$, where
340 $t \geq k$. To this end, we define $B' = \frac{1}{m}\sum_{i\in[m]}\left(\widehat{A}_t^{(i)} - \widehat{A}_k^{(i)}\right)$. By definition, we have $B' = \widetilde{A}_t - \widetilde{A}_k$.
341 Now, let us relate $B'$ and $\widetilde{B}_k$. Note that for any machine, $\widehat{A}_t^{(i)} - \widehat{A}_k^{(i)} \preceq \widehat{A}^{(i)} - \widehat{A}_k^{(i)}$, by definition.
342 Thus by taking averages, we have that $B' \preceq \widetilde{B}_k$.

343 We will now argue that with probability $\geq 1 - \delta$,

$$B' = \Pi^\perp B'\Pi^\perp + E, \quad \text{where } \|E\| \leq O(\kappa)\log(1/\delta). \tag{7}$$

344 To see this, let us expand the first term on the RHS using $\Pi^\perp = (I - \Pi)$:

$$\Pi^\perp B'\Pi^\perp = B' - B'\Pi - \Pi B' + \Pi B'\Pi.$$

345 Now, since $B' \preceq \widetilde{B}_k$, we have $\|B'\Pi\| \leq \|\widetilde{B}_k\Pi\| \leq O(\kappa)\log(1/\delta)$, by the earlier claim. Thus the
346 last three terms are all bounded in norm by $O(\kappa)\log(1/\delta)$, and hence we have the desired bound on
347 $\|E\|$.

348 Putting (6) and (7) together, we have that with probability at least $1 - \delta$,

$$\widetilde{A}_t = \widetilde{A}_k + B' = A_k + \Pi^\perp B'\Pi^\perp + E', \text{ where } \|E'\| \leq O(\kappa)\log(1/\delta).$$

349 Now, because $A_k$ and $\Pi^\perp B'\Pi^\perp$ are in orthogonal spaces, the eigenvalues of $A_k + \Pi^\perp B'\Pi^\perp$ are
350 precisely the union of the eigenvalues of the two matrices. The eigenvalues of $A_k$ are simply

351    $\lambda_1, \ldots, \lambda_k$. We claim that $\lambda_{\max}(B') \leq \lambda_{k+1} + O(\kappa) \log(1/\delta)$, with probability at least $(1 - \delta)$.
352    This can be shown as follows. First, since $B' \preceq \widetilde{B}_k$, it suffices to bound $\lambda_{\max}(\widetilde{B}_k)$. Since $\widetilde{B}_k =$
353    $(A - A_k) + (A_k - \widetilde{A}_k) - (A - \widetilde{A})$, using (5) and (6), it follows that with probability at least $1 - \delta$,

$$\lambda_{\max}(\widetilde{B}_k) \leq \lambda_{\max}(A - A_k) + O(\kappa \log(1/\delta)) = \lambda_{k+1} + O(\kappa \log(1/\delta)).$$

354    Thus, due to the gap between $\lambda_k$ and $\lambda_{k+1}$, the top $k$ eigenvalues of $A_k + \Pi^\perp B' \Pi^\perp$ are exactly
355    $\lambda_1, \ldots, \lambda_k$. Thus by Weyl's inequality, the eigenvalues of $\widetilde{A}_t$ satisfy (3). This completes the
356    proof.                                                  □

## Footnotes

[3]For any matrices $X, Y$, $\|XY\|_F \leq \|X\|_F\|Y\|_2$. (This is easy to show, by observing how $Y$ acts on the rows of $X$.)