[Reviews · NeurIPS 2019]

Reviewer 1



========================================================= After response, I maintain my evaluation. I agree the technical contribution might be a bit incremental comparing with previous works, but I do see the value of this paper. I think it would also be good if this paper is getting accepted. ========================================================= Although results look solid and quite interesting compared to previous work, I'm a bit concerned with the writing qualities of this paper. Main theorem (Theorem 1) and Corollary 2 and Corollary 3 are stated without settings/assumptions/preconditions. The statement of Theorem 1 has a notion of subGaussian norm, which is claimed to be described in the preliminaries while this paper does not has a preliminary. (I eventually realize this notion is defined in footnote 1 without mentioning its name...) Section 2.1 is almost just filled with theorems and corollaries without too much explanations. Even though section 1.2 informally stated the results and their implications, there is still a big gap when stating the formal theorems without much explanations. In sum, I think the results are interesting, and have reasonable significance especially when compared to [6]. However, I feel authors could possibly polish the paper a bit more. The current version looks a bit rough.

Reviewer 2



****** post rebuttal ****** I have read the authors response. I'm not sure I'm convinced regarding the level of novelty of the arguments used, nevertheless I think this is a solid and nice contribution and I would like to keep my score. ***************************** The authors revisit the problem of distributed stochastic k-pca, where m machines each have an i.i.d. sample of size n. The authors consider a one-shot aggregation protocol, where each machine computes locally on her sample and sends O(kd)-size summary to a central node. The central node then aggregates these individual summaries to extract an estimate for the top-k components of the covariance of the underlying (unknown) distribution, with the aim of achieving statistical accuracy as if these estimates were directly computed from the entire m*n sample of all machines combined. While previous works [6,7] focused only on extracting the top k-dimensional subspace (i.e., an orthonormal basis for the span of the top k eigenvectors of the underlying covariance), here the authors consider estimating the exact top-k decomposition, including the eigenvalues and eigenvectors (if they are unique, otherwise some basis to corresponding eigenspace). While [6] proposed that each machine sends its top-k eigenvectors computed over the sample (to estimate only the subspace), here the authors consider a natural extension, in which each machine sends the eigenvectors, each scaled by square-root of the corresponding eigenvalue. Sinne, as in previous works, the estimation accuracy depends on the eigengap between the k and k+1 largest eigenvalues, the authors also show a procedure that using rank of k_1 they can extract the decomposition for any k<=k_1 depending on the kth-eigengap. This is useful when we do not know exactly for which k the gap is large enough. In terms of technique, the authors rely heavily on results established in [6]. In this aspect, the work is somewhat incremental and I do not identify highly novel ingredients. It would be great if the authors can emphasize the technical difficulties and novel solutions. Overall this is a nice result that has a solid contribution and, to my taste, worthy of publication in NIPS

Reviewer 3



The algorithm differs from [6] in that instead of averaging the actual eigenvectors, it averages the eigenvectors weighted by the corresponding eigenvalues. The claim is that this modification enables a better concentration result in terms of estimating each eigenvector. A small notational issue -- it might be useful to state upfront that when used without a subscript (which sometimes has been done), "\| \|" refers to the spectral(?) norm. The "bias" lemma is quite interesting, and this is clearly the technically most challenging part of the paper. It would be very illustrative if the authors could present a clearer intuition of what "cancellations" enable a better bound here. I was wondering why the authors do not refer to any of the randomized sketching methods for EVD/SVD? It seems to me that there is a feasible algorithm in which each of the machines either sketches the samples and then combines them to find the actual SVD. It also seems to me that as long as we get a guarantee corresponding to Theorem 1, it is possible to get bounds on individual eigenvalue/vector estimation similar to Corollary 4 & 5. It would be nice to at least compare the results, both theoretical guarantees as well as empirically, with these methods.

Reviewer 4



I still think the contribution is incremental so I maintain my score. ------------------------------------------------------ The new algorithm let each machine sends the square-root of the local covariance matrix to the server in contrast to previous works that let each machine sends eigenvectors. This new method does make sense because sending only eigenvectors will create additional bias during the local estimation step. The analysis and proof in the current draft do reflect it. Furthermore, empirical experiments do verify it. I think this paper makes contributions to the distributed PCA problem. However, I feel the contribution is limited as it only corrects the sub-optimal estimators in previous work.

[Author Response · NeurIPS 2019]

We thank the reviewers for the valuable comments and suggested improvements.

**Reviewer 1: Theorems in Section 2.1 stated without reference to motivation.** In the final version, we will interleave the theorem statements from Section 2.1 and Section 4 with the corresponding discussion in Section 1.2. This will hopefully aid the flow and add value to the remarks and discussion in Section 1.2. The reference to the preliminaries will be replaced with one to Section 1.1 (Problem setup and motivation).

**Reviewer 2: Technical challenges compared to [6].** The works [6] and [7] developed bounds for how the top-k eigenspace is perturbed under sampling and aggregation. In our algorithm, we need to analyze how the *low rank approximation* is perturbed. While the techniques are inspired by those works, we extended (and simplified) their results for our purposes. We will highlight the differences further in the final version. Finally, the analysis for imprecise $k$ (Theorem 9) is entirely novel; indeed the setting itself has not been studied in prior works. Not knowing the precise location of an eigenvalue-gap is quite natural in applications, and thus we view this as an important setting to study.

**Reviewer 5: Intuition about cancellations in Theorem 4.** We thank the reviewer for pointing this out. The explanation for the linear (dominant in magnitude) terms cancelling currently appears in the proof at the end of page 5. We will include the main idea in the proof overview at the start of the section.

**Reviewer 5: Comparison to work on sketching for EVD/SVD.** The works based on sketching have been crucial to our understanding of distributed SVD. We will include a discussion on these techniques in the final version. A key difference between the setting we consider and many of those works is that we focus on finding the precise eigenvectors/values, as opposed to minimizing the low rank error. The two notions are related in many realistic settings, and it is an interesting direction to compare them. At a high level, averaging methods (such as ours) perform really well as the number of points per machine (parameter $n$) grows. In contrast, sketching methods perform better as the sketch size grows (while not being too dependent on $n$).

**Reviewer 6: Contribution and tightness of the bounds.** Our results may indeed be viewed as correcting the estimators from prior work. Indeed, the algorithm itself changes only slightly. However, this change leads to significant improvements both theoretically and empirically for the fundamental problem of estimating the SVD. Further, the analysis now involves more work, as described in the response to review #2 (please see above).

The bounds in the theorems are tight as far as the dependence on the parameters $m, n$ and the gaps ($\Delta_k$ terms) go. It is possible that dependence on the trace can be improved. We will include a detailed discussion (and relevant open questions) in the final version. We note that optimality in terms of $n$ (i.e., the first term in Theorem 1) was shown in [7], and the dependence on $\sqrt{mn}$ follows from the Sin-theta theorem and known optimality results for matrix concentration.

[Meta-Review · NeurIPS 2019]

The reviewers agree that, while the technical ingredients are somewhat incremental over the prior work in [6], this paper presents a nice contribution to the distributed PCA problem. Several reviewers highlighted ways in which presentation of the paper can be polished -- we ask the authors to take these suggestions into consideration when editing the paper.